# A Tale of 12 Tails: Katanin Severing Activity Affected by Carboxy-Terminal Tail Sequences

**DOI:** 10.3390/biom13040620

**Published:** 2023-03-30

**Authors:** K. Alice Lindsay, Nedine Abdelhamid, Shehani Kahawatte, Ruxandra I. Dima, Dan L. Sackett, Tara M. Finegan, Jennifer L. Ross

**Affiliations:** 1Physics Department, Syracuse University, New York, NY 13244, USA; 2Department of Chemistry, University of Cincinnati, Cincinnati, OH 45221, USA; 3*Eunice Kennedy Shriver* National Institute of Child Health and Human Development, NIH, Bethesda, MD 20892, USA

**Keywords:** microtubule-severing enzyme, tubulin isotypes, tubulin code, post-translational modifications, katanin, microtubule-associated protein

## Abstract

In cells, microtubule location, length, and dynamics are regulated by a host of microtubule-associated proteins and enzymes that read where to bind and act based on the microtubule “tubulin code,” which is predominantly encoded in the tubulin carboxy-terminal tail (CTT). Katanin is a highly conserved AAA ATPase enzyme that binds to the tubulin CTTs to remove dimers and sever microtubules. We have previously demonstrated that short CTT peptides are able to inhibit katanin severing. Here, we examine the effects of CTT sequences on this inhibition activity. Specifically, we examine CTT sequences found in nature, alpha1A (TUBA1A), detyrosinated alpha1A, Δ2 alpha1A, beta5 (TUBB/TUBB5), beta2a (TUBB2A), beta3 (TUBB3), and beta4b (TUBB4b). We find that these natural CTTs have distinct abilities to inhibit, most noticeably beta3 CTT cannot inhibit katanin. Two non-native CTT tail constructs are also unable to inhibit, despite having 94% sequence identity with alpha1 or beta5 sequences. Surprisingly, we demonstrate that poly-E and poly-D peptides are capable of inhibiting katanin significantly. An analysis of the hydrophobicity of the CTT constructs indicates that more hydrophobic polypeptides are less inhibitory than more polar polypeptides. These experiments not only demonstrate inhibition, but also likely interaction and targeting of katanin to these various CTTs when they are part of a polymerized microtubule filament.

## 1. Introduction

Microtubules are essential cytoskeletal structures crucial for cell division, maintaining cellular structure, and intracellular transport. Microtubules are rigid, hollow fibers formed by the polymerization of dimers of two closely related globular proteins: alpha- and beta-tubulin. Microtubules are highly dynamic and intrinsically grow and shrink in a manner that is known as dynamic instability (reviewed extensively including [1]). Microtubule properties and dynamics are intrinsically differentially regulated by the incorporation of different tubulin isotypes and post-translational modifications. This is known as the ‘tubulin code’ and allows for specialization of microtubules for different functions (reviewed extensively, including [2,3,4,5]). Microtubule dynamics are also regulated extrinsically by a host of microtubule-associated proteins (MAPs) [6]. MAPs respond to the ‘tubulin code’, which influences their affinity for binding.

Notable amongst MAPs that regulate microtubules is katanin, a conserved AAA ATPase MAP complex that severs and destabilizes microtubules, and consequently regulates cellular homeostasis in a number of different cellular pathways (reviewed in [7,8]). Dysregulation of animal katanin is implicated in developmental, proliferative, and neurodegenerative disorders (reviewed in [9]). Katanin is composed of a 60 kD catalytic subunit (p60), with ATPase and microtubule-severing activities, and a regulatory 80 kD subunit (p80) that regulates the subcellular localization of the complex [10]. Different oligomerization states of katanin are found [11,12,13] but like most AAA ATPase enzymes, katanin p60 is active with respect to microtubule severing when in a hexameric ring complex bound to microtubules [14,15]. ATP binding induces hexamerization of the AAA domain and tubulin hydrolysis causes disassembly of the ring complex [10,14]. Human katanin has a poor propensity for oligomerization, which is speculated to prevent aberrant activation and conformation [12].

Katanin regulates microtubule length by both severing and promoting depolymerization [16,17,18,19]. Katanin binding and severing on microtubule filaments are separate mechanisms [18,19]. Katanin microtubule regulation is inhibited by: (1) sequestration of p60 enzymatic monomer units such that it cannot bind microtubules and (2) posttranslational modifications to p60 [11,18,20,21,22,23].

We have previously addressed how microtubules are regulated by katanin by modifying the in vitro microtubule severing assay developed [24] to quantify binding of a green fluorescent protein-labeled katanin p60 construct (GFP-katanin) and subsequent severing [18]. Using this assay, we previous demonstrated that adding free tubulin into this assay inhibits the severing action of katanin through competitive and irreversible binding to katanin p60 monomers [18]. Further, short, synthesized peptides with CTT sequences showed the same effect. We have previously found that beta tails are more efficient at inhibiting katanin p60 severing of microtubules in this assay, and therefore that beta CTT sequences more efficiently bind katanin p60.

Here, we examine CTT sequences found in nature, namely alpha1A (TUBA1A), detyrosinated alpha1A, Δ2 alpha1A, beta5 (formerly called TUBB [25], recently called TUBB5 [26]), beta2a (TUBB2A), beta3 (TUBB3), and beta4b (TUBB4b) [26]. The sequences, charge distributions, and peptide shapes are similar with stretches of negatively charged glutamic acid residues separated by glycine residues (Table 1). The incorporation of hydrophobic residues affects the hydrophobicity of the constructs, as measured using the Wimley–White scale (Table 1) [27]. We find that natural CTTs have distinct abilities to inhibit katanin severing in vitro. As previously shown, alpha1A CTT is better at inhibiting than detyrosinated alpha1A CTT, yet Δ2 alpha1A CTT is better than detyrosinated alpha1A CTT. While beta5 CTT is still the most potent inhibitor of the beta tails, and beta2a and beta4b CTTs are both capable of inhibiting, beta3 CTT is incapable of inhibiting severing by katanin.

These results indicate that the last amino acids are important to the binding and inhibition of CTTs to katanin, so we created two artificial constructs: an alpha1A CTT with the last amino acid (Y) replaced with a phenylalanine and a beta5 CTT construct with the last amino acid (A) replaced with a tyrosine (Table 1). Neither of these two artificial CTT constructs were able to inhibit, despite having 94% sequence identity with alpha1 or beta5 sequences. Finally, we examined the role of polyglutamylation and charge on the ability to inhibit katanin using constructs with 10 glutamic acids or 10 aspartic acids. Surprisingly, both the poly-E and poly-D peptides are capable of inhibiting katanin significantly. Although this assay examines inhibition, this inhibition is likely caused by competitive binding of the CTTs for the katanin. Thus, these assays provide evidence about which ‘tubulin codes’ can regulate katanin activity in cells.

## 2. Materials and Methods

The methods used to purify GFP-katanin and perform the in vitro assays and quantify the data have been previously published [18,19].

Peptides corresponding to the CTT sequences of human tubulins, and modified forms, were custom synthesized and obtained from Peptide 2.0 (Chantilly, VA, USA). Purity and sequence accuracy were confirmed by mass spectrometry performed by Dr. Lisa Jenkins, NCI, NIH.

Complete methods can be found in Appendix B.

## 3. Results

In this work, we use the in vitro microtubule severing assay to quantitatively characterize the microtubule binding and severing ability of human GFP-katanin p60 in the presence of different CTT sequences [18]. First, we ensure that our human GFP-katanin p60 can robustly and reproducibly sever microtubules and completely remove all polymer within 3–5 min (Figure 1A). As each katanin prep has slightly different activity and the activity can fluctuate (Appendix A), we perform control experiments each day to determine the optimal katanin concentration needed and to serve as a control without added inhibitors (Figure 1A). For simplicity we use “katanin” to refer to human GFP-katanin p60 for the remainder of this manuscript.

We quantify the intensity of the loss of microtubule polymer and normalize the data to start at 100% (Figure 1(Aii)), as previously performed [18]. We also quantify the GFP-katanin binding and normalize the data so that the maximum GFP intensity is one when there are no inhibitors present (Figure 1(Aiii)). As expected, katanin is an ATPase, so it requires ATP to sever, which we observe (Figure 1(Bi)). Without ATP, the loss of polymer is slow (Figure 1(Bii)) and appears to be due to depolymerization from filament ends as opposed to severing (Figure 1(Bi)). We find that GFP-katanin is still able to bind slowly to microtubules even in the absence of ATP (Figure 1(Bi,Biii)). These data are normalized to the same day control with katanin and ATP without inhibitor (Figure 1A) and demonstrates that 70% less GFP-katanin is able to bind in the absence of ATP (Figure 1(Biii)).

Katanin has a higher affinity for free tubulin than microtubules and unlike spastin, katanin is inhibited by the presence of tubulin dimers [18,28,29]. We recapitulate that activity (Figure 1C) and demonstrate that the tubulin dimers inhibit severing and depolymerization (Figure 1(Ci,Cii)) which is a consequence of the lack of binding (Figure 1(Ci,Ciii)).

Our previous work using this assay showed that katanin’s microtubule severing activity is (1) dependent on katanin p60 concentration; (2) inhibited by free tubulin dimers; and (3) inhibited specifically and differentially by the CTTs of tubulin. Here, we use this robust assay to quantitatively interrogate the ability of different tubulin CTT sequences to bind and sequester katanin, and consequently inhibit katanin-microtubule binding and microtubule severing to reveal how the biochemical and biophysical properties of native and non-native CTT sequences, part of the ‘tubulin code,’ influence katanin-mediated microtubule severing.

### 3.1. Katanin Severing Inhibition by Whole Tubulin Dimers, Alpha1a, and Beta5 CTTs

Previously, we showed that free tubulin dimers, alpha1a tubulin CTTs, and beta5 tubulin CTTs can each inhibit katanin severing of microtubules, but there are quantitative differences in the dynamics of katanin binding and rate of microtubule severing [18]. In that prior study, the concentration of CTT was held constant. Here, we vary the amount of alpha1a (TUBA1A/B) or beta5 (TUBB5) CTT from 0 to 2 μM and compare the effects of varying additional tubulin dimers over the same range. We find that, as previously shown, whole tubulin dimers are capable of inhibiting the binding and the severing of microtubules by katanin in a concentration-dependent manner (Figure 2(Ai,Bi)). At the highest concentrations of free tubulin (1.5, 2 μM), the amount of katanin binding is not visible above the background level (Figure 2(Bi)). Further, at a high concentration of free tubulin, the percentage of microtubule polymer over time exhibits a characteristic linear decrease, showing that the loss of polymer is mostly from depolymerization of the filaments—not rapid severing as observed at low tubulin dimer concentrations (Figure 2(Ai), best fits given in Appendix A).

We also showed that beta5 tubulin CTT peptides were better at inhibiting katanin than whole tubulin, while alpha1a tubulin CTTs were not as potent at inhibiting [18]. Indeed, we observe that trend again here, especially at high concentrations of alpha1 and beta5 CTTs (Figure 2(Aii,Aiii), best fits given in Appendix A) where the characteristic decay times are longer at lower beta5 CTT concentrations (Figure 2C). Further, for concentrations where the microtubule polymer is lost, the GFP signal is high as katanin rapidly binds to the microtubules, and then low, due to the loss of polymer to which the katanin binds (Figure 2(Bii,D)).

Interestingly, for both the beta5 and the alpha1a CTTs at high concentration, the severing is inhibited and initial binding rates for GFP-katanin are significantly lower, but the intensity of GFP-katanin on the filaments increases slowly over time and can reach levels as high as in the absence of CTTs (Figure 2(Bii,Biii)). Specifically, this can be seen in the GFP katanin plots for alpha1a CTTs at 2 μM (Figure 2(Bii)) and beta5 CTTs at 0.5 and 1 μM (Figure 2(Biii)).

Qualitatively, it appears that the intensity on the individual microtubules does increase, especially near regions where the microtubule is depolymerizing, as seen in the representative kymograph showing GFP-katanin binding to a microtubule in the presence of 2 μM alpha1a CTT (Figure 2(Ei)). This is similar to what is observed in the absence of ATP (Figure 1B). Quantification of the intensity of GFP-katanin along the microtubule at early, middle, and late times shows that the intensity increases even as some regions reach background levels of intensity due to loss of polymer from the ends or sparse severing events in the middle of the filament (Figure 2(Eii)). Even though there is loss of polymer, the average intensity of GFP-katanin increases, especially compared to the background level measurement, that only shows minor photobleaching (Figure 2(Eiii)). Only at the very end, when the microtubule is depolymerizing is there any intensity reduction for the example shown (Figure 2(Eiii)), and not all filaments had the same depolymerization in the field (see Appendix A, Alpha1a, 2 μM for the movie).

We previously observed a similar accumulation of katanin at the ends of depolymerizing microtubules lacking the CTT [19]. It is unclear if the accumulation of katanin at tips causes microtubule depolymerization or vice versa. It could be that the depolymerization of the filaments causes the accumulation of higher concentrations of loosely bound katanin to the ends. Alternatively, it is also possible that the higher accumulation of katanin locally causes the depolymerization of the microtubules. The inhibition caused by high levels of alpha1a and beta5 CTT peptides reduces the rate of katanin binding, while the katanin is still capable of binding and triggering some severing and possibly some depolymerization.

The phenomena of slow accumulation of GFP-katanin without severing was not observed for whole tubulin dimers. Further, previous studies that used CTTs attached to BSA also did not see this accumulation [18]. One major difference between the current study and the previous one is we are using small peptides, while the previous study created constructs that were BSA globular protein with multiple CTTs covalently attached. Further, tubulin dimers also have a globular body attached to the CTTs. The globular domain (BSA or tubulin) could increase the inhibition of the katanin binding and jam the pore of the katanin oligomers, making it difficult for the katanin to bind the microtubules. Conversely, the short CTT peptides we use here that could easily translocate through an oligomer pore, freeing a subset of katanin hexamers to be able to bind to the microtubules.

Taken together, it appears that GFP-katanin binding results in loss of microtubule polymer. Thus, high GFP-katanin binding causes fast severing rates. We calculated the rate from the characteristic decay time using: r=1/τ and plotted the severing rate as a function of the measured average, maximum GFP value (Figure 2F). Apart from the outliers that are high GFP-katanin due to long-time accumulation in the inhibited experiments (denoted in the dashed box), we see a linear relationship between the GFP bound to the microtubules and the rate of loss of polymer. The lines are best fit lines where the accumulation data were excluded to perform the fit (see Appendix A best fit parameters).

### 3.2. Katanin Inhibition by Alpha Tail Modifications

A major post-translational modification of tubulin is the detyrosination of the alpha CTT where the final amino acid of the alpha tail, a tyrosine, is removed [2,30,31]. Detyrosinated microtubules can be found among wholly unmodified filaments within the cell. Detyrosination is associated with increased stability to cold treatment in cells [32], but not in vitro [33] and reduced binding of destabilizing motor proteins [34]. Further removal of the next amino acid, a glutamic acid, results in a post-translational modification called Δ2 (Table 1).

We created alpha1a tail constructs with these two modifications and tested their ability to inhibit katanin binding and severing. Both detyrosinated (alpha1a-Y) and Δ2 (alpha1a-YE) tails were capable of inhibiting katanin, but not as well as full-length alpha1a tubulin CTTs at the highest concentrations (Figure 3(Ai,Aii), fit parameters in Appendix A). Characteristic times increase with increasing CTTs for all with the same trends (Figure 3C). We quantified the GFP-katanin binding over time (Figure 3(Bi,Bii)), and found the same trend—full length alpha CTTs inhibited the binding of katanin best, then Δ2 (alpha-YE) and finally detyrosinated (alpha-Y), which is clear in the data showing the maximum average GFP on the microtubules (Figure 3D). Based on the data, it appears that alpha CTT is the most active inhibitor, Δ2 (alpha1a-YE) CTT is the next best, and detyrosinated alpha (alpha1a-Y) CTT is the least effective of the three, although they are all capability of inhibiting binding and severing of katanin at the highest concentrations of CTTs.

Using the GFP-katanin intensity data, we determined the rate for GFP-katanin to associate to the microtubules after being flowed into the chamber (Figure 3D). The binding rate is slowest for alpha1a CTTs and the fastest for detyrosinated CTTs with Δ2 CTTs between (Figure 3E). The binding rate decrease occurs for all constructs, but it is most apparent for the detyrosinated CTTs. Similar to some of the data observed for alpha1a and beta5 CTTs, the Δ2 CTTs at the highest concentration (2 μM) showed the slow binding and leveling off that was observed with alpha CTTs (Figure 3(Bii)). The movies revealed the same slow severing and depolymerization observed for alpha1a CTTs (see Appendix A).

Finally, we plot the severing rate (inverse of the characteristic severing time) to the maximum amount of GFP bound to the filament (Figure 3F). Again, we see a linear dependence implying that the severing is directly proportional to the amount of katanin that is able to bind. The boxed region has the data that had a slow GFP-katanin binding rate, but resulted in a high intensity at longer times. These data were excluded from the fit (fit parameters given in Appendix A).

Overall, the effects of the removal of the last few amino acids from the alpha1 tubulin CTT have quantitative and qualitative effects, but the alpha1a peptide is still able to inhibit the binding and severing activity of katanin to similar degrees.

### 3.3. Katanin Inhibition by Beta Tail Isotypes

Beta tubulin isotype CTTs are more diverse in their sequence differences compared to alpha isotype CTTs (Maleikal 2022). Here, we examine some of the most diverse tails to determine if they have different abilities to inhibit katanin. Specifically, we examine the tail sequences of beta2a (TUBB2A), beta3 (TUBB3), and beta4b (TUBB4B), which we compare to beta5 (TUBB5) (Table 1).

Examining beta2a CTTs, the sequence is not so dissimilar from beta5 CTTs (Table 1). The most striking difference is the disruption of a chain of repeating glutamic acids with a glycine and the addition of a glutamate and a glycine closer to the amino-terminus of the peptide. These subtle differences are enough to shift the inhibition concentration, so that more beta2a CTT is needed to inhibit katanin binding and severing compared to beta5 CTTs (Figure 4(Ai,C), best fits are given in Appendix A). Along with the loss of severing is the same reduction in GFP-katanin binding (Figure 4(Bi,D)).

The beta3 CTTs are the most unusual in terms of sequence, with various unique residues at the end ending with a positively charged lysine. We find that beta3 CTTs cannot inhibit katanin severing (Figure 4(Aii), best fits are given in Appendix A). Further, there is little inhibition of GFP-katanin binding (Figure 4(Bii)). Despite the quantitative reduction in severing rate and binding, for the two highest concentrations of beta3 CTT (1.5 and 2 μM), the movies qualitatively show clear and rapid severing of microtubules even at 2 μM beta3 CTT (see Appendix A). Beta3 CTTs cannot inhibit katanin binding or severing activity.

Beta4b CTT’s sequence is more canonical than beta3 CTTs except the presence of a valine with the alanine at the very end of the peptide. These residues are uncharged and hydrophobic, which could affect interactions with katanin. We find that beta4b CTTs are able to inhibit katanin severing (Figure 4(Aiii), best fits are given in Appendix A) and binding (Figure 4(Biii)), although to a lesser extent than beta or beta2a CTTs (Figure 4A,C). Linear fits to the severing rate plotted against the maximum GFP intensity show the same linear dependence (Figure 4E, best fits given in Appendix A).

The sequence of the CTT appears to have a significant impact with the peptides ability to interact with and inhibit katanin. From the beta CTT tail data, it appears that relatively small changes that are naturally occurring can have large impacts on interactions (Figure 4). Interestingly, for both the alpha and beta tails tested, changes to the end of the CTT construct have impacts on katanin interactions (Figure 3 and Figure 4).

### 3.4. Examination of Katanin Inhibition by Artificial CTTs Where the Last Amino Acid Is Altered

As demonstrated above for the alpha and beta CTTs, the tail sequence—especially the last amino acid—can impact the interaction and inhibition activity on katanin’s binding and severing ability. Here, we investigate that effect by creating two artificial tail constructs, one based on alpha1a and one based on beta5 CTTs, to determine the impacts of a change just to the very c-terminus of the peptide sequence. The two constructs we created are named alpha1a-Y+F CTT and beta5-A+Y CTT.

For the alpha1a-Y+F CTT, removing the final tyrosine and adding a phenylalanine residue to the very end of the alpha CTT should not significantly change the size or charge of the peptide, since tyrosine and phenylalanine are only different by the addition of an OH group on the carbon ring. Yet, the changing of the final amino acid on alpha1a CTT has major consequences to katanin binding and severing. The alpha1a-Y+F CTT has no inhibitory effect on GFP-katanin severing (Figure 5(Ai), best fits given in Appendix A) or binding (Figure 5(Bi). This one seemingly minor change demonstrates the importance of the final residue to the activity of the alpha1a CTTs in their interaction with katanin. The severing rates appear linear with respect to the GFP-katanin binding (Figure 5E, see Appendix A for best fits).

The alpha1a CTT data implies that the final amino acid being a tyrosine is very important for GFP-katanin inhibition. The beta5 CTT is the most potent inhibitor of the CTTs we tested. In order to determine if the tyrosine will enhance the inhibition caused by beta5 CTTs, we created an artificial beta5-A+Y construct where we remove the final alanine and replace it with a tyrosine. Surprisingly, we find that this construct is far less potent as a katanin inhibitor (Figure 5(Aii), best fits given in Appendix A) and does not affect katanin binding (Figure 5(Bii)). This result again confirms the importance of the final residue and indicates that the tyrosine is not optimized for the beta5 CTT’s interaction with katanin, and instead completely eliminates the interaction.

### 3.5. Katanin Inhibition by Poly-E and Poly-D Peptides

One major similarity between alpha and beta CTTs is that there are repeats of glutamic acids within the sequence for both. Further, there is an additional post-translational modification that can occur on both the alpha and beta CTTs of polyglutamylation, where multiple branched poly-E peptides can be attached to residue 445 of alpha1a or beta2A tubulin or beta3 tubulin at residue 435 and 438, respectively [2,5,23]. Thus, poly-E peptides may alter the interactions of the tubulin CCTs with associated enzymes and proteins.

To test if poly-E peptides can interact with and inhibit katanin, we created a non-native CTT sequence composed of 10 glutamic acid residues in a row (10E). We also made a 10 amino acid sequence of aspartic acid (D10) to test amino acid specificity using another reside that is also charged. The D10 construct has the same charge as the E10, but has a slightly modified structure. Although no poly-aspartic acid chains are post-translationally modified on tubulin, there are regions of the naturally occurring alpha and beta CTTs that have aspartic acids instead of glutamic acids.

We find that both the E10 and the D10 sequences can inhibit the severing activity (Figure 6A, best fits given in Appendix A) and binding of GFP-katanin (Figure 6B). Interestingly, despite having no specific CTT sequence, the E10 and D10 inhibition activity follows a similar trend in inhibition of characteristic times, specifically, they inhibit best at the higher concentrations (Figure 6C, 1.5 and 2 μM). These two concentrations also have significantly reduced katanin binding rates and maximum binding intensities (Figure 6D). The linear dependence on severing rate as a function of GFP-katanin binding is better for D10 compared to E10 (Figure 6E, best fits given in Appendix A). The fact that poly-E and poly-D peptides can inhibit katanin severing is surprising considering that these constructs are not CTTs. They do not have the interspersed uncharged residues and are not ended with a hydrophobic residue as the other constructs are. The katanin inhibition of these two constructs imply that negative charge is a likely driver for the interaction with katanin.

## 4. Discussion

We used an in vitro microtubule severing assay we developed previously with purified GFP-katanin to interrogate if the CTT sequence of tubulin regulates inhibition of katanin microtubule severing [18]. These studies are important to understand the ability of different microtubule substrates to be severed by katanin. Although we are using an inhibition assay, the inhibition is mediated by the ability of the severing enzyme to bind to the CTT introduced. If there is significant inhibition by a CTT sequence, this indicates that there is a good interaction of katanin for that sequence—good enough to compete with the microtubule substrate present. For sequences that are good inhibitors, we interpret that the same sequence, when part of a tubulin dimer incorporated into a microtubule, will also be a good substrate for katanin. Thus, all the good inhibitors are likely good targets for katanin degradation. Poor inhibitors are likely poor targets for katanin interaction and degradation.

A recent study has shown that microtubules with post-translational modifications do display different abilities for katanin binding and severing [23]. Many of the trends we report here for inhibition are also observed for microtubules made with post-translationally modified tubulins. Specifically, they observe that poly-glutamylation of alpha tails results in more binding and severing whereas polyglutamylation of beta tails is biphasic with more binding, but non-monotonic increase than decrease in severing activity [23]. Our results with poly-E peptides also predict the same increased binding interaction they observe. They also find that detyrosination of the alpha tails results in decreased binding and severing activity [23], as we also observe with our CTTs.

We find that the carboxy tails of alpha, beta, modified alphas (detyrosinated and Δ2), and some beta isotypes are moderate to good targets for katanin. Perhaps more intriguing are the poor targets. Beta3 CTT is a poor target for katanin. Beta3 isoforms are found in brain nerve cells—specifically expressed in the axons and dendrites [35]. Mutants of beta3 tubulin result in impairment of vision and large-scale brain structure deformations that can result in neurological defects [36]. It also appears to be important for nerve cell regrowth [37]. It is interesting that such an important isoform is not a target for katanin. It is known that katanin defects also result in poor development of the brain—specifically causing microcephaly and lissencephaly [38,39,40]. Interestingly, there is no overlap in the structures or phenotypes between katanin mutations or deletion phenotypes and beta3 deletion or mutation phenotypes—implying that they likely do not interact with each other, despite being both important required proteins for brain development.

Perhaps surprisingly, detyrosinated alpha1a CTT is still capable of inhibiting katanin, implying it is a moderate good target for katanin degradation. Detyrosination is associated with stability of microtubules in cells [32]. Yet, in vitro reconstitution of detyrosinated microtubules shows that stability is not conferred directly from the detyrosination [33]. Thus, it is most likely that detyrosination is a signal to destabilizing elements to avoid these microtubules. It has been shown that detyrosinated microtubules are avoided by MCAK, a depolymerizing kinesin [34]. Here, we show some reduced affinity of katanin with detyrosinated alpha1a CTTs, but not so significant to imply it could not be degraded by katanin.

The other constructs that were poor targets were the non-native peptides we made from alpha1a and beta5 CTT sequences. These constructs demonstrate that the importance of the last amino acid in the sequence for both the alpha and beta sequences. It was surprising that the phenylalanine had such a drastic effect, given the similarities between it and tyrosine. Further, if tyrosine is so important, why is the beta tail sequence with a tyrosine attached as the last residue such a poor inhibitor? To add to our surprise, we found that two constructs with poly-E or poly-D inhibited katanin just as well as native tails. The E10 construct is like polyglutamylation post-translational modification that can occur on alpha or beta tails.

We have found that katanin activity is sensitive to the pH of the solution. Specifically, katanin is completely inhibited at pH 6.6 (Appendix A). It prefers a higher pH closer to 8, and we use pH 7.7 in our assays because microtubules prefer a slightly acidic buffer (PEM is a PIPES buffer with pH 6.8). If the addition of the E10 or the D10 constructs could change the pH, that could be the cause of the inhibition. We calculate that, for the highest concentration of D10 or E10, the pH would only shift by 0.01—not enough to cause the deactivation of katanin due to pH. Thus, we believe that inhibition is truly caused by the D10 and E10 peptides interacting with katanin directly. Future works examining the interaction using other techniques may reveal the binding affinity of these unnatural tail sequences.

An interesting aspect of the CTTs is that they are all negatively charged (Table 1), but apart from being negative, there does not seem to be a correlation between their activity and their overall charge. For instance, the beta3 CTT has a total charge of −11, but it is a poor inhibitor of katanin. Instead, we focus on another biochemical and biophysical aspect of the CTTs—their hydrophobicity. Based on the recent findings from the literature [41], we evaluated the hydrophobicity of the peptides using the Wimley–White scale [27], which represents the free energy change when transferring pentapeptides between water and a lipid bilayer. This scale was shown to perform very well in several experimental and computational tests, compared to the more traditional scales, such as the Kyte–Doolitle scale [41]. For each of the 11 peptides tested in our work, we calculated their average hydrophobicity (the total hydrophobicity of the peptide divided by its length) in order to compare directly peptides of different lengths (Table 1).

As discussed above and in the literature [33], alpha and beta-tubulin tails interact differently with katanin. Alpha-tails have a preferred interaction outside of the AAA core of the motor, while the beta-tails interact directly with the ATPase domain by contacting the pore loops of the protomers in the hexameric states of the motor. We used the data from Table 1 to rank order the alpha peptides and the beta peptides separately.

We found that the synthetic alpha1a-Y+F peptide is the most hydrophobic of the five alpha tails, followed by the alpha1a CTT, and alpha1-YE CTT, with alpha1-Y CTT being the most hydrophilic. For the beta tails, we found that beta3 CTT is the most hydrophobic peptide, followed by beta5-A+Y CTT, beta2a CTT, beta5 CTT, and finally beta4 CTT. The 2 acidic peptides, D10 and E10 are highly polar, with E10 being the most polar of the 11 peptides.

Although the hydrophobicity trends do not exactly replicate the trends we observe, there is a much better correlation than the total charge. As example, for both alpha and beta tails, the most hydrophobic peptides are unable to inhibit katanin severing. In contrast, more polar peptides inhibit severing by katanin. Namely, the two synthetic peptides built from the alpha1a and the beta5 CTTs are more hydrophobic than the corresponding wild-type tails. Beta3 CTTs is the most hydrophobic of all beta CTTs. These are the three tubulin tails that are unable to inhibit severing. Even for beta2a CTT, which is more hydrophobic than beta5 CTT, we found that it needs higher concentration, compared to beta5 CTT, to inhibit severing. We also note that beta5 CTT is a better inhibitor of katanin severing than alpha1a CTT, which correlates with the fact that alpha1a CTT is more hydrophobic than beta5 CTT. For more hydrophilic peptides, such as alpha1a-Y CTT, we found a decrease in katanin microtubule binding and severing, which has been reported also by a recent study [23], indicating that more hydrophilic tails are good inhibitors of katanin binding and severing of microtubules. Finally, we note that among the human alpha tubulins, alpha8, which plays a critical role in cortical progenitor differentiation [33], is the only tail with a C-terminal F. This tail has the sequence DSFEEENEGEEF resulting in an average hydrophobicity of −1.20, which makes it the most hydrophobic of all the tails. Based on our results, especially the synthetic alpha1a-Y+F tail, we predict that the presence of phenylalanine at the end of an alpha-tail increases its hydrophobicity, compared to the wild-type alpha1a CTT, and will make it unable to inhibit katanin severing.

Analysis of the hydrophobicity of the peptides shows that for both alpha and beta CTTs, any change that increases the average hydrophobicity of the peptide leads to a reduced or complete loss in the ability to inhibit katanin severing. In addition, peptides characterized by high hydrophilicity inhibit severing more readily than hydrophobic ones, which explains the inhibitory effect of the poly-D and poly-E tails. We note that the results of the hydrophobicity analysis are robust even at the more basic pH used in our experiments, as the differences between the behavior of the peptide tails are due to the amino acid types that have been shown to have hydrophobicity values independent of pH over a large interval [42].

## 5. Conclusions

The new experimental results described here support the tubulin code model for how MAPs and microtubule-enzymes can be directed to act in time and space by the sequence and post-translational modifications to tubulin’s CTT. Future screens of this sort can be used to make predictions about which tubulin isotypes are good or poor targets for degradation by katanin. Katanin activity has also been linked to downstream increasing of microtubule networks and enhanced stability of individual filaments [29,43,44]. Thus, these experiments will also predict which tissues expressing different isotypes might have less dense microtubule networks or less stable filaments. Future in vitro experiments with different tails, modified tubulins as microtubules, and even expressing different isotypes in live cells will be needed to address the predictions from this work.

## Figures and Tables

**Figure 1 biomolecules-13-00620-f001:**
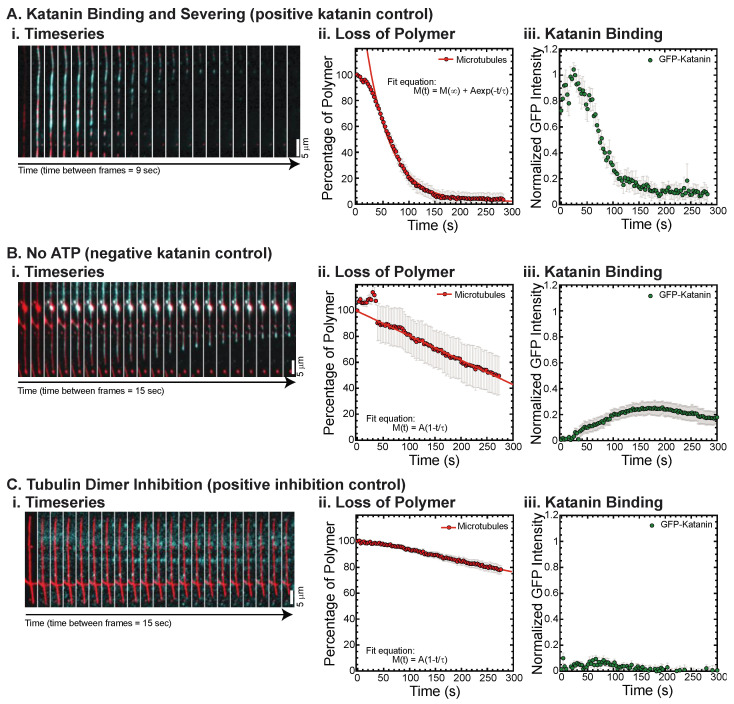
Example data, quantification, and controls. (**A**) Katanin in the presence of ATP is able to bind and sever microtubules. This daily control demonstrated that katanin was active. (**i**) Time series of a representative movie showing GFP-katanin binding (cyan) along rhodamine-labeled microtubule (red), becoming severed and losing polymer throughout the filament over time. Scale bar is 5 μm. Time between consecutive frames is 9 s. (**ii**) Quantification of the loss of polymer measured the signal on the microtubule and the noise 5 pixels away to determine the S/N-1 and then normalized by the initial time data. Loss of polymer data was fit with an exponential decay equation. Best fit parameters for these data are given in Appendix A. (**iii**) GFP-katanin binding was quantified in the same way and the S/N-1 was normalized compared to the same day control with ATP (these data), which is why these data have a maximum of 1. (**B**) Katanin in the absence of ATP is able to bind, although not as well. This daily control demonstrated that katanin ATPase activity was responsible for severing. (**i**) Time series of a representative movie showing GFP-katanin binding (cyan) along rhodamine-labeled microtubule (red), not severing, but losing polymer from the ends, depolymerizing. Scale bar is 5 μm. Time between consecutive frames is 15 s. (**ii**) Quantification of the loss of polymer as in part (**A**). Loss of polymer data was fit with a linear approximation to an exponential decay equation. Best fit parameters for this data are given in Appendix A. (**iii**) GFP-katanin binding was quantified in the same way and the S/N-1 was normalized compared to the same day control with ATP. (**C**) Katanin in the presence of ATP and 2 μM tubulin dimers, which inhibit binding and severing. (**i**) Time series of a representative movie showing GFP-katanin does not bind well (cyan) along rhodamine-labeled microtubule (red). Scale bar is 5 μm. Time between consecutive frames is 15 s. (**ii**) Quantification of the loss of polymer as in part (A). Loss of polymer data was fit with a linear approximation to an exponential decay equation. Best fit parameters for these data are given in Appendix A. (**iii**) GFP-katanin binding was quantified in the same way and the S/N-1 was normalized compared to the same day control with ATP. Representative movies in Appendix A.

**Figure 2 biomolecules-13-00620-f002:**
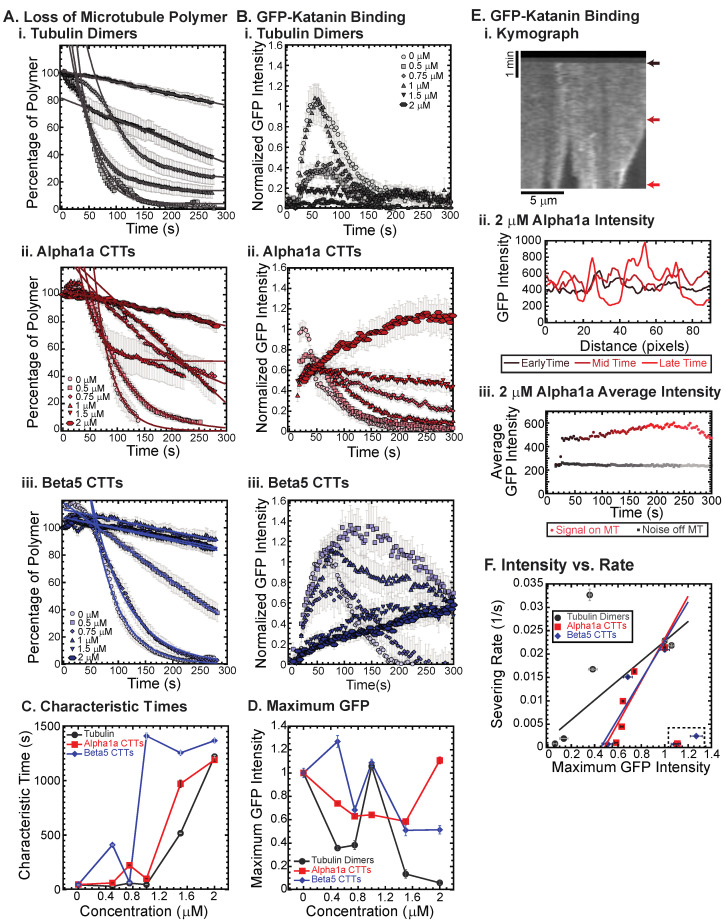
Inhibition of katanin binding and severing by tubulin dimers, alpha CTTs, and beta CTTs. (**A**) Loss of polymer caused by increasing concentration of (**i**) tubulin dimers (gray), (**ii**) alpha CTTs (red), and (**iii**) beta CTTs (blue). The transition from exponential decay to linear decay is observed for loss of polymer data as it goes from no inhibitor (0 μM, lightest shade circles), low inhibitor (0.5 μM, light shade squares; 0.75 μM, light shade diamonds), medium inhibitor (1 μM, medium shade triangles with point up), to highest inhibitor (1.5 μM, darker shade triangles with point down, 2 μM, darkest shade hexagons). Each dataset is fit with a best fit equation to an exponential or linear decay with best fit parameters provided in the Appendix A. (**B**) The data for the GFP-katanin binding by increasing concentration of (**i**) tubulin dimers (gray), (**ii**) alpha CTTs (red), and (**iii**) beta CTTs (blue). The data were normalized by the maximum when there was no inhibitor (0 μM, lightest shade circles). Data for low inhibitor (0.5 μM, light shade squares; 0.75 μM, light shade diamonds), medium inhibitor (1 μM, medium shade triangles with point up), to highest inhibitor (1.5 μM, darker shade triangles with point down, 2 μM, darkest shade hexagons) are shown. (**C**) The characteristic decay times for loss of polymer as a function of concentration for tubulin (gray circles), alpha1 CTTs (red squares), and beta CTTs (blue diamonds). Error bars represent the uncertainty in the fit parameters. (**D**) The average normalized maximum for GFP-katanin binding to microtubules for tubulin (gray circles), alpha1 CTTs (red squares), and beta CTTs (blue diamonds). Error bars represent the standard deviation of the average. (**E**) Representative data for GFP-katanin binding in the presence of 2 μM alpha1 CTT demonstrates the accumulation of GFP-katanin over time is not an artifact. (**i**) Kymograph of a representative microtubule showing GFP-katanin binding along the microtubule length (horizontal direction, scale bar is 5 μm of microtubule length) over time (vertical direction, scale bar is 1 min of movie data). Arrows indicate the times used in part (**ii**). (**ii**) Intensity scans over the microtubule length at early (black line), middle (maroon line), and late (red line) time points indicated in part (**i**) show that the intensity fluctuations have increased due to loss of polymer (lowest intensities) and accumulation of GFP-katanin (highest intensities). (**iii**) Intensity averaged over the length of the microtubule shown in part (**i**) plotted over time of the movie for the same region of interest on the microtubule (red circles) and off the microtubule (gray squares) shows that the background intensity is not increasing. (**F**) Plotting the rate of microtubule severing as a function of average normalized maximum intensity for tubulin (gray circles), alpha1 CTTs (red squares), and beta CTTs (blue diamonds). Data with high GFP-katanin intensity due to a slow accumulation with little loss of polymer denoted by dashed line box (lower right). Removing these data, we fit the severing rate to a linear fit as a function of GFP-katanin. Best fit parameters given in Appendix A. Representative movies in Appendix A.

**Figure 3 biomolecules-13-00620-f003:**
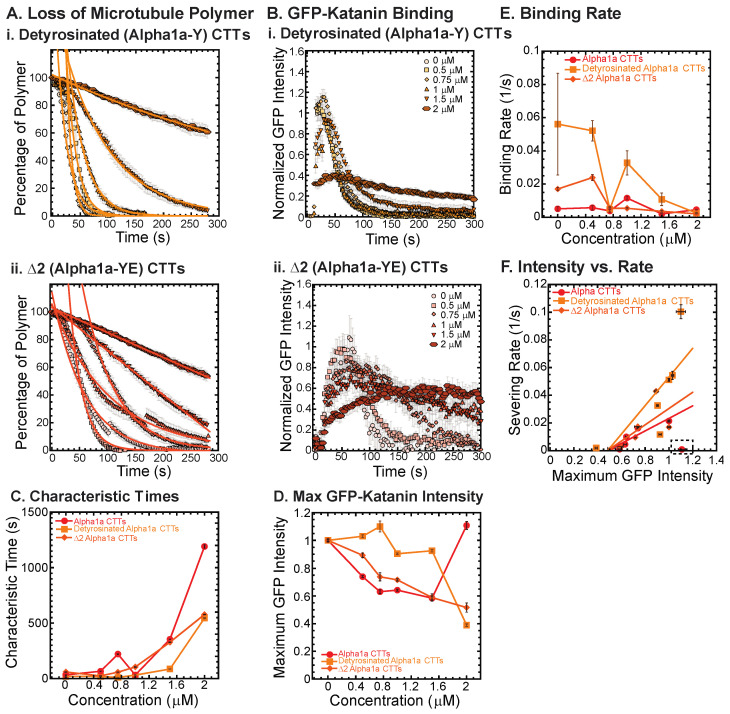
Effects of alpha1a CTT post-translational modifications on katanin inhibition. (**A**) Loss of polymer caused by increasing concentration of (**i**) detyrosinated alpha1a CTTs (light orange) and (**ii**) Δ2 alpha1a CTTs (dark orange). The transition from exponential decay to linear decay is observed for loss of polymer data as it goes from no inhibitor (0 μM, lightest shade circles), low inhibitor (0.5 μM, light shade squares; 0.75 μM, light shade diamonds), medium inhibitor (1 μM, medium shade triangles with point up), to highest inhibitor (1.5 μM, darker shade triangles with point down, 2 μM, darkest shade hexagons). Each dataset is fit with a best fit equation to an exponential or linear decay with best fit parameters provided in Appendix A. (**B**) The data for the GFP-katanin binding by increasing concentration of (**i**) detyrosinated alpha1a CTTs (light orange) and (**ii**) Δ2 alpha1a CTTs (dark orange). The data were normalized by the maximum when there was no inhibitor (0 μM, lightest shade circles). Data for low inhibitor (0.5 μM, light shade squares; 0.75 μM, light shade diamonds), medium inhibitor (1 μM, medium shade triangles with point up), to highest inhibitor (1.5 μM, darker shade triangles with point down, 2 μM, darkest shade hexagons) are shown. (**C**) The characteristic decay times for loss of polymer as a function of concentration for alpha1a CTTs (red circles), detyrosinated alpha1a CTTs (light orange squares) and Δ2 alpha1a CTTs (dark orange diamonds). Error bars represent the uncertainty in the fit parameters. (**D**) The average normalized maximum for GFP-katanin binding to microtubules for alpha1a CTTs (red circles), detyrosinated alpha1a CTTs (light orange squares) and Δ2 alpha1a CTTs (dark orange diamonds). Error bars represent the standard deviation of the average. (**E**) GFP-katanin binding was measured for the initial time points for each data set by fitting a linear fit equation from the initial times to the time when maximum is reached. The slopes of the fits represent the rate of katanin binding, plotted as a function of CTT concentration for alpha1a CTTs (red circles), detyrosinated alpha1a CTTs (light orange squares) and Δ2 alpha1a CTTs (dark orange diamonds). Error bars represent the uncertainty in the fit parameters. (**F**) Plotting the rate of microtubule severing as a function of average normalized maximum intensity for alpha1a CTTs (red circles), detyrosinated alpha1a CTTs (light orange squares) and Δ2 alpha1a CTTs (dark orange diamonds). Data with high GFP-katanin intensity due to a slow accumulation with little loss of polymer denoted by dashed line box (lower right). Removing this data, we fit the severing rate to a linear fit as a function of GFP-katanin, provided in Appendix A. Representative movies in Appendix A.

**Figure 4 biomolecules-13-00620-f004:**
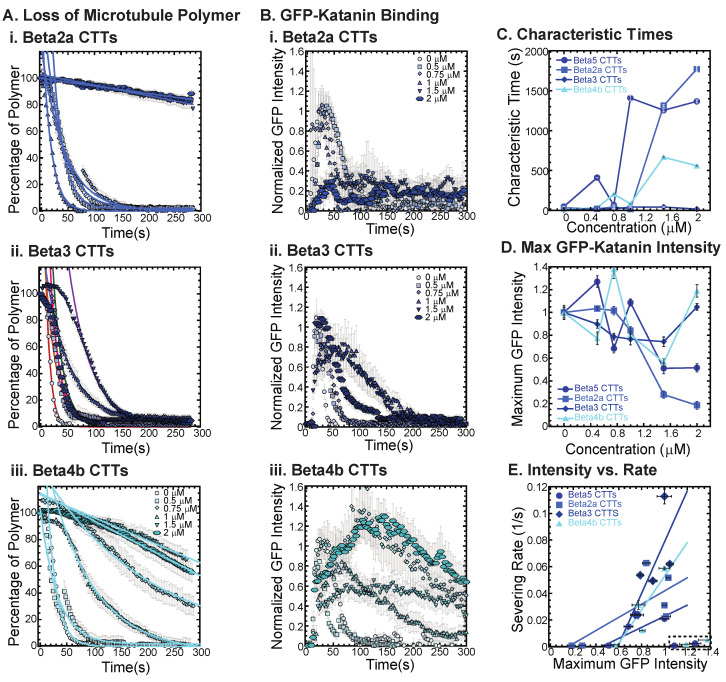
Effects of beta tubulin isotype sequence on katanin inhibition. (**A**) Loss of polymer caused by increasing concentration of (**i**) beta2a CTTs (light blue), (**ii**) beta3 CTTs (navy blue), and (**iii**) beta4b CTTs (cyan). The transition from exponential decay to linear decay is observed for loss of polymer data as it goes from no inhibitor (0 μM, lightest shade circles), low inhibitor (0.5 μM, light shade squares; 0.75 μM, light shade diamonds), medium inhibitor (1 μM, medium shade triangles with point up), to highest inhibitor (1.5 μM, darker shade triangles with point down, 2 μM, darkest shade hexagons). Each dataset is fit with a best fit equation to an exponential or linear decay with best fit parameters provided in Appendix A. (**B**) The data for the GFP-katanin binding by increasing concentration of (**i**) beta2a CTTs (light blue), (**ii**) beta3 CTTs (navy blue), and (**iii**) beta4b CTTs (cyan). The data were normalized by the maximum when there was no inhibitor (0 μM, lightest shade circles). Data for low inhibitor (0.5 μM, light shade squares; 0.75 μM, light shade diamonds), medium inhibitor (1 μM, medium shade triangles with point up), to highest inhibitor (1.5 μM, darker shade triangles with point down, 2 μM, darkest shade hexagons) are shown. (**C**) The characteristic decay times for loss of polymer as a function of concentration for beta CTTs (bright blue circles), beta2a CTTs (light blue squares), beta3 CTTs (navy blue diamonds), and beta4b CTTs (cyan triangles). Error bars represent the uncertainty in the fit parameters. (**D**) The average normalized maximum for GFP-katanin binding to microtubules for beta CTTs (bright blue circles), beta2a CTTs (light blue squares), beta3 CTTs (navy blue diamonds), and beta4b CTTs (cyan triangles). Error bars represent the standard deviation of the average. (**E**) Plotting the rate of microtubule severing as a function of average normalized maximum intensity for beta CTTs (bright blue circles), beta2a CTTs (light blue squares), beta3 CTTs (navy blue diamonds), and beta4b CTTs (cyan triangles). Data with high GFP-katanin intensity due to a slow accumulation with little loss of polymer denoted by dashed line box (lower right). Removing these data, we fit the severing rate to a linear fit as a function of GFP-katanin. Representative movies in Appendix A.

**Figure 5 biomolecules-13-00620-f005:**
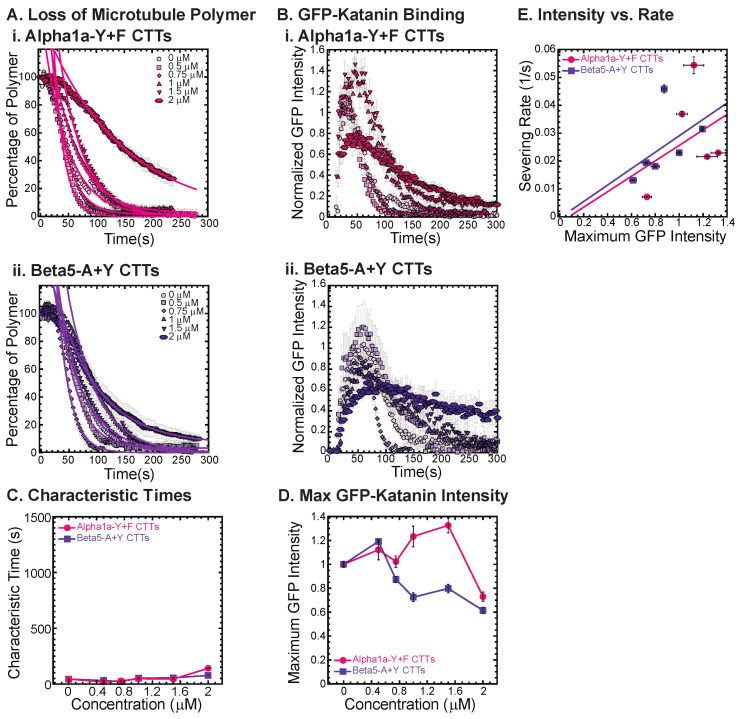
Katanin severing and binding not inhibited by artificially constructed CTTs. (**A**) Loss of polymer caused by increasing concentration of (**i**) alpha-Y+F CTTs (magenta) and (**ii**) beta-A+Y CTTs (purple). All datasets were fit to exponential decays because these CTTs did not inhibit severing significantly as the amount of inhibitor is changed from no inhibitor (0 μM, lightest shade circles), low inhibitor (0.5 μM, light shade squares; 0.75 μM, light shade diamonds), medium inhibitor (1 μM, medium shade triangles with point up), to highest inhibitor (1.5 μM, darker shade triangles with point down, 2 μM, darkest shade hexagons). Each dataset is fit with a best fit equation to an exponential with best fit parameters provided in Appendix A. (**B**) The data for the GFP-katanin binding by increasing concentration of (**i**) alpha-Y+F CTTs (magenta) and (**ii**) beta-A+Y CTTs (purple). The data were normalized by the maximum when there was no inhibitor (0 μM, lightest shade circles). Data for low inhibitor (0.5 μM, light shade squares; 0.75 μM, light shade diamonds), medium inhibitor (1 μM, medium shade triangles with point up), to highest inhibitor (1.5 μM, darker shade triangles with point down, 2 μM, darkest shade hexagons) are shown. (**C**) The characteristic decay times for loss of polymer as a function of concentration for alpha-Y+F (magenta circles) and beta-A+Y CTTs (purple squares). Error bars represent the uncertainty in the fit parameters. (**D**) The average normalized maximum for GFP-katanin binding to microtubules for alpha-Y+F (magenta circles) and beta-A+Y CTTs (purple squares). Error bars represent the standard deviation of the average. (**E**) Plotting the rate of microtubule severing as a function of average normalized maximum intensity for alpha-Y+F (magenta circles) and beta-A+Y CTTs (purple squares). No data displayed high GFP-katanin intensity due to a slow accumulation with little loss of polymer denoted by dashed line box (lower right). All data of the severing rate were fit to a linear function of GFP-katanin (see Appendix A for best fits). Representative movies in Appendix A.

**Figure 6 biomolecules-13-00620-f006:**
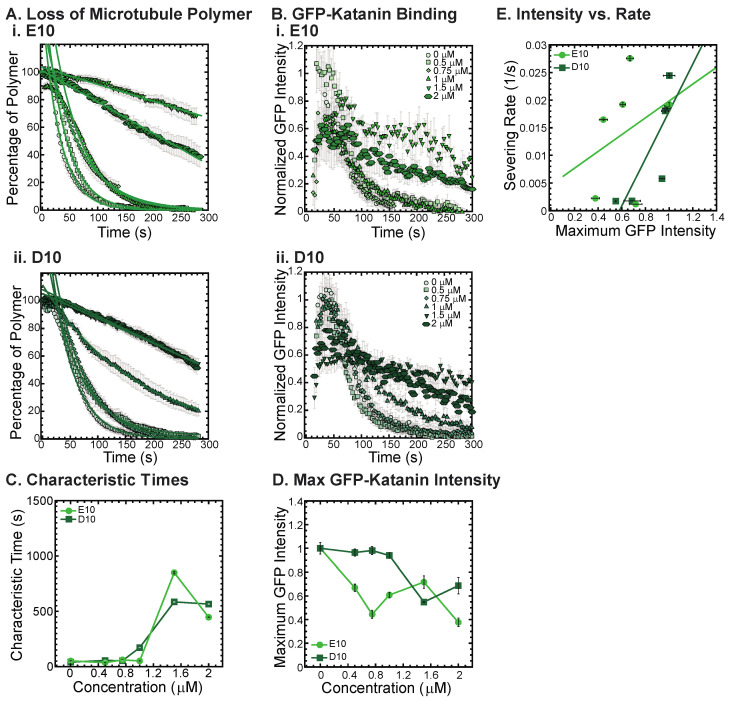
Katanin severing and binding inhibited by poly-E and poly-D peptides. (**A**) Loss of polymer caused by increasing concentration of (**i**) E10 peptides (light green) and (**ii**) D10 peptides (dark green). Datasets were fit to exponential decays because these CTTs did not inhibit severing significantly as the amount of inhibitor is changed from no inhibitor (0 μM, lightest shade circles), low inhibitor (0.5 μM, light shade squares; 0.75 μM, light shade diamonds), medium inhibitor (1 μM, medium shade triangles with point up), to highest inhibitor (1.5 μM, darker shade triangles with point down, 2 μM, darkest shade hexagons). The transition from exponential decay to linear decay is observed for loss of polymer data as it goes from no inhibitor (0 μM, lightest shade circles), low inhibitor (0.5 μM, light shade squares; 0.75 μM, light shade diamonds), medium inhibitor (1 μM, medium shade triangles with point up), to highest inhibitor (1.5 μM, darker shade triangles with point down, 2 μM, darkest shade hexagons). Each dataset is fit with a best fit equation to an exponential or linear decay with best fit parameters provided in Appendix A. (**B**) The data for the GFP-katanin binding by increasing concentration of (**i**) E10 peptides (light green) and (**ii**) D10 peptides (dark green). The data were normalized by the maximum when there was no inhibitor (0 μM, lightest shade circles). Data for low inhibitor (0.5 μM, light shade squares; 0.75 μM, light shade diamonds), medium inhibitor (1 μM, medium shade triangles with point up), to highest inhibitor (1.5 μM, darker shade triangles with point down, 2 μM, darkest shade hexagons) are shown. (**C**) The characteristic decay times for loss of polymer as a function of concentration for E10 peptides (light green circles) and D10 peptides (dark green squares). Error bars represent the uncertainty in the fit parameters. (**D**) The average normalized maximum for GFP-katanin binding to microtubules for E10 peptides (light green circles) and D10 peptides (dark green squares). Error bars represent the standard deviation of the average. (**E**) Plotting the rate of microtubule severing as a function of average normalized maximum intensity for E10 peptides (light green circles) and D10 peptides (dark green squares). No data displayed high GFP-katanin intensity due to a slow accumulation with little loss of polymer denoted by dashed line box (lower right). All data of the severing rate was fit to a linear function of GFP-katanin (see Appendix A). Representative movies in Appendix A.

**Table 1 biomolecules-13-00620-t001:** CTT constructs used in this paper showing the name, sequence, total charge, hydrophobicity, and structure. Charge was determined from summing the positive and negative amino acids at pH 7.7. Hydrophobicity was determined using the Wimley-White scale, where a lower number is more hydrophobic and a higher number is less hydrophobic.

Name	Sequence	Charge	Hydrophobicity	Structure
Alpha1a(TUBA1A/B)	** A T A D S V E G E G EEE G EE Y **	−8	−1.30	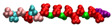
Alpha1-Y(detyrosinated)	** A T A D S V E G E G EEE G EE **	−8	−1.49	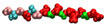
Alpha1-YE(Δ2)	** A T A D S V E G E G EEE G E **	−7	−1.40	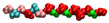
Beta5(TUNN/TUBB5)	** A T A EEEED F G EE A EEE A **	−10	−1.58	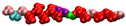
Beta2(TUBB2A)	** A T A DE Q G E F EEEE G EDE A **	−10	−1.51	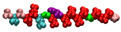
Beta3(TUBB3)	** A T A EEEED M Y EDDDEE S E A Q G P K **	−11	−1.28	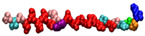
Beta4b(TUBB4B)	** A T A EE G E F EEE A EE VA **	−10	−1.61	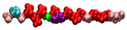
Alpha-Y+F	** A T A D S V E G E G EEE G EE F **	−8	−1.28	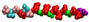
Beta5-A+Y	** A T A EEEED F G EE A EEE Y **	−10	−1.48	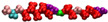
E10	** A T A EEEEEEEEEE **	−10	−2.51	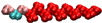
D10	** A T A DDDDDDDDDD **	−10	−1.53	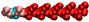

**Red**: Negatively charged; **Cyan**: Polar uncharged; **Purple**: Hydrophobic aromatic ring; **Blue**: Positively charged; **Pink**: Hydrophobic; **Green** & **Orange**: Special cases.

## Data Availability

Microscopy data reported in this paper will be shared by the corresponding author upon request. This paper does not use any original code.

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
