# Peer review of "A Tale of 12 Tails: Katanin Severing Activity Affected by Carboxy-Terminal Tail Sequences"

_biomolecules, 2023, doi:10.3390/biom13040620_

Round 1
Reviewer 1 Report
This proposal examines effects of amino acid sequences of the tubulin carboxy-terminal tail (CTT) on the inhibition of effects of AAA Atpase katanine on tubulin. The authors show extensive set of data. Since this reviewer is not quite familiar with the field, I am unable to comment on technical aspects of experimental procedures. Although the paper seems to be suitable for publication, the following points may need to be attended (some are significant).
Lines 82-83: An awkward sentence “Neither of these two artificial CTT con- 82 structs were unable to inhibit – despite having 94% sequence identity with alpha1 or beta5 83 sequences.” I don’t understand what is meant here. Do the authors mean neither of these constructs able to inhibit?
Since katanin activity is unstable, please give how much activity is lost per unit time and at what level of activity loss was the cutoff for the usage of the batch? This is significant because if the inactive (partially unfolded?) protein content is high it may impact the binding measurements (e.g. nonspecific binding). What type of correction may be applied?
In Figure 2 (and most others), it is difficult to follow shades of the same color because of significantly overlapping curves. Figures presented are usually “too busy”. They may need to be simplified by separating and or eliminating some. Specifically, since similar assays were used , some may simply be moved to supplementary materials. This may also eliminate the excessive length of the figure legends.
I am also having a difficult time to understand variations on GFP intensity in Figure 2D. Why is the signal going up and down? (I don’t think that the explanation given on lines 184-186 are clear and sufficient).
Author Response
Response to reviewers’ comments:
Review 1:
This proposal examines effects of amino acid sequences of the tubulin carboxy-terminal tail (CTT) on the inhibition of effects of AAA Atpase katanine on tubulin. The authors show extensive set of data. Since this reviewer is not quite familiar with the field, I am unable to comment on technical aspects of experimental procedures. Although the paper seems to be suitable for publication, the following points may need to be attended (some are significant).
We appreciate that the reviewer thinks that the paper seems acceptable for publication.
Lines 82-83: An awkward sentence “Neither of these two artificial CTT con- 82 structs were unable to inhibit – despite having 94% sequence identity with alpha1 or beta5 83 sequences.” I don’t understand what is meant here. Do the authors mean neither of these constructs able to inhibit?
We apologize for the very confusing typo, and thank the reviewer for finding it in their careful reading of the manuscript. The sentence should say that neither were ABLE to inhibit. We have corrected it, and we thank the reviewer for their help.
Since katanin activity is unstable, please give how much activity is lost per unit time and at what level of activity loss was the cutoff for the usage of the batch? This is significant because if the inactive (partially unfolded?) protein content is high it may impact the binding measurements (e.g. nonspecific binding). What type of correction may be applied?
This is a good question. We have performed a series of severing assays over three weeks and found a minimal amount of loss of function, but larger than expected fluctuations. We still believe that it is best to perform same-day controls to compare inhibition experiments to the activity without inhibitors on the same day. We have updated the language on lines 111-114 to discuss this. We have reported the activity of the same protein preparation over 21 days in a new supplemental figure S1 and report all the fit parameters in new supplemental table S12.
In Figure 2 (and most others), it is difficult to follow shades of the same color because of significantly overlapping curves. Figures presented are usually “too busy”. They may need to be simplified by separating and or eliminating some. Specifically, since similar assays were used , some may simply be moved to supplementary materials. This may also eliminate the excessive length of the figure legends.
We would like to keep the data in the same figures, but we understand about the readability. That being said, given that we give all the fit data, it will be possible for readers to understand the trends. Finally, we are happy to share the raw data with anyone who would like to redo the analysis.
I am also having a difficult time to understand variations on GFP intensity in Figure 2D. Why is the signal going up and down? (I don’t think that the explanation given on lines 184-186 are clear and sufficient).
We have repeated the data to measure the GFP intensity for 0, 1000, 2000. We actually find a similar result, so we have opted to keep the original data set. We do not have a good explanation for the result apart from some fluctuations that could occur from assay to assay. We instead focus on the overall trends and comparisons between different CTT constructs.
Reviewer 2 Report
This is a very nice paper describing different inhibitory roles of various tubulin C-terminal tails (CTTs) to the microtubule-severing protein Katanin. I strongly support its publication and only have two very minor points for the authors to consider.
1. In the abstract and main text, the authors should add "CTT" after the name of a particular tubulin when its inhibitory role was described. For example, in the abstract, "beta3 cannot inhibit katanin" sounds confusing and should be changed to "beta3 CTT cannot inhibit katanin".
2. In appendix A, it would be better to add more details about the synthesized peptides (for example, are they modified?) if possible.
Author Response
Reviewer 2:
This is a very nice paper describing different inhibitory roles of various tubulin C-terminal tails (CTTs) to the microtubule-severing protein Katanin. I strongly support its publication and only have two very minor points for the authors to consider.
We thank the reviewer for this positive assessment.
- In the abstract and main text, the authors should add "CTT" after the name of a particular tubulin when its inhibitory role was described. For example, in the abstract, "beta3 cannot inhibit katanin" sounds confusing and should be changed to "beta3 CTT cannot inhibit katanin".
Thank you for this clarifying remark. We agree and have added CTT in the appropriate places.
- In appendix A, it would be better to add more details about the synthesized peptides (for example, are they modified?) if possible.
The peptides were chemically synthesized by a company that specializes in peptide synthesis (Peptide 2.0, Chantilly VA). Each peptide was examined by mass spectrometry after receipt to assure that it was full length (i.e. corresponded to the full sequence mass) and was not modified (i.e. demonstrate that they did not have mass excess over the theoretical sequence mass). We have added this last bit to clarify for the reader. Thank you for pointing out that it was not clear.